# Exploring the Origin and Physiological Significance of DNA Double Strand Breaks in the Developing Neuroretina

**DOI:** 10.3390/ijms23126449

**Published:** 2022-06-09

**Authors:** Noemí Álvarez-Lindo, Teresa Suárez, Enrique J. de la Rosa

**Affiliations:** 3D Lab (Development, Differentiation and Degeneration), Centro de Investigaciones Biológicas Margarita Salas, CSIC, Ramiro de Maeztu 9, 28040 Madrid, Spain; alvarezlnl@cib.csic.es

**Keywords:** DSBs, neuron somatic mosaicism, Rag2, recombination, NHEJ, neurogenesis

## Abstract

Genetic mosaicism is an intriguing physiological feature of the mammalian brain that generates altered genetic information and provides cellular, and prospectively functional, diversity in a manner similar to that of the immune system. However, both its origin and its physiological significance remain poorly characterized. Most, if not all, cases of somatic mosaicism require prior generation and repair of DNA double strand breaks (DSBs). The relationship between DSB generation, neurogenesis, and early neuronal cell death revealed by our studies in the developing retina provides new perspectives on the different mechanisms that contribute to DNA rearrangements in the developing brain. Here, we speculate on the physiological significance of these findings.

## 1. Introduction

In the last decade, new single cell sequencing technologies have completely changed our understanding of neural DNA, revealing that almost all healthy individuals studied carry large numbers of neuron-specific genetic alterations, most of which require prior generation and repair of DNA double strand breaks (DSBs) [1,2,3,4,5,6,7,8]. This variability is far more frequent than ever expected. Single cell genomics has demonstrated somatic mosaicism in physiological contexts in more than 10% of neurons within a given individual, increasing to 90% in some studies [1,2,3,4]. These findings indicate that cell heterogeneity in the central nervous system (CNS) relies not only on transcriptional, morphological and functional diversity, but also on major, likely underlying, changes in neuronal DNA. 

Neuronal DNA is enriched as a consequence of multiple genetic alterations in neural progenitor cells. Affected regions range in size from over 500 Mb to single nucleotides (single nucleotides variations, SNVs) (Figure 1). These major genetic alterations include indels (insertion/deletion), MEIs (mobile element insertions), CNVs (copy number variations), SVs (structural variants), and aneuploidy (see [9]). Advances in single-cell whole-genome sequencing have provided detailed information about smaller CNVs (<1 Mb) and have shown that these are very frequent in the developing cerebral cortex [10]. All these forms of somatic mosaicism require prior generation and repair of DNA DSBs, which in turn must be properly repaired to prevent programmed cell death of the affected neuronal cell.

This somatic mosaicism within neurons results in changes in their gene expression [11], although these changes do not necessarily affect neuronal connectivity or survival in the adult brain [12]. Particularly, aneuploid neurons have been found to be active and fully integrated in the normal adult mammalian brain [12,13]. Moreover, the recent discovery of abundant CNVs of <1 Mb revealed quantitative variation at particular developmental stages in the mouse cerebral cortex [10], suggesting that these alterations may constitute part of a process intrinsic to neural development.

**Figure 1 ijms-23-06449-f001:**
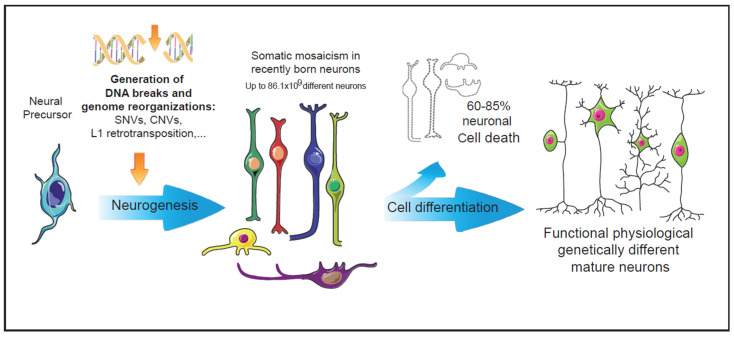
Intrinsic events during early neurogenesis that may contribute to somatic mosaicism and functional diversity in the mature nervous system. In the nervous system, neuronal genetic diversity seems to arise intrinsically during early neuronal differentiation. Diverse genetic alterations have been observed in healthy neurons, most of them involving the generation and repair of DNA double strand breaks. The pool of neurons with cell-unique differences in their DNA could be even larger than the vast numbers of antibodies generated by the V(d)J recombination in the immune system [14]. During neuronal differentiation, more than 60% of the recently generated neurons undergo cell death events [15]. The surviving neurons, many of them carrying genetic alterations, may configure a functional repertoire characterized by the physiologically generated somatic mosaicism.

Recurrent DSBs are also a feature of many psychiatric and neurodegenerative diseases [16,17], and somatic mosaicism appears to play a role in the physiopathology of brain diseases [18]. However, the role of genetic variations in normal neural development and its possible impact on brain diseases remains unclear, since the mechanisms underlying the generation of DSBs and consequent somatic mosaicism in the brain remain largely unexplored.

## 2. DSBs and Neural Development

DSBs can result from extrinsic causes, including certain viruses, ionizing radiation, and chemical sources, or from intrinsic causes, such as reactive oxygen species produced by cellular respiration or replication fork collapse at genome fragile sites [19,20,21,22]. Alternatively, they can be a consequence of specific mechanisms and examples include programmed genome reorganization, such as RAG-1,2 endonuclease-mediated V(D)J somatic recombination, as it occurs in the immune system [23,24]. In the brain, DSBs can also be formed by specific mechanisms, such as retrotransposon mobilization (e.g., LINE-1 transposition in neural progenitors and mature neurons [25,26]) and RNA retroinsertion, as in somatic APP gene recombination [27]. Specific DSB generation mediated by endonucleases to control gene expression, as described for DNA topoisomerase II β (TOP2β) and Spo11 in the promotors of neuronal activity-induced genes [28,29,30], have also been shown. 

DSBs constitute a potentially serious threat to cell survival and, therefore, must be properly repaired. DSBs are repaired by either homologous recombination or nonhomologous end-joining (NHEJ). NHEJ involves the direct ligation of the two DNA ends that frequently alter the original DNA sequence. A defective DNA damage response severely impacts nervous system development [31], and alterations in factors involved in the DSB response have been implicated in a range of diverse human syndromes, including neuropathology and neurodegeneration (e.g., ataxia telangiectasia) (Table 1) [32,33,34]. The importance of DSBs and their consequences for neurons are clearly evidenced by the phenotype of mice carrying DNA repair mutations. These mice present a marked, and in some cases lethal, embryonic phenotype, characterized by a high level of neuronal cell death, impaired development, and even acellularity in the central nervous system [15,35] (see Table 1). Indeed, mouse models with NHEJ mutations (e.g., XRCC4 and DNA lig IV) display a dramatic neural phenotype, in some cases with embryonic lethality [36,37], and severe immunodeficiency (Table 1), although the remaining organs and tissues are largely unaffected, suggesting an important role of NHEJ in neural development.

DSBs occur in proliferative areas of the CNS but, in contrast to that described in most other cell types, DSBs do not necessarily stop the cell cycle during neuronal development, nor trigger immediate cell apoptosis [65]. In good agreement with those findings, recurrent DSB clusters have also been detected in neural stem or progenitor cells from the frontal brain of mice [66], and recent mouse studies have estimated that the onset of frequent DSB-dependent CNVs in all chromosomes occurs at E13-E14, coinciding with the neurogenic process [10]. Our own findings in mouse retina also indicate an increase in developmental DSBs at E13-E14 [48], an effect that is even more pronounced in DNA repair mutants [48,62].

Together, these data suggest that the neurogenic process provides an environment permissive to DSB generation and subsequent genetic alterations. The specificity of these neural DNA alterations is supported by the observation that genetic changes emerge specifically during in vitro stem cell differentiation towards neural lineage [3] but are completely absent during differentiation towards fibroblasts. This accumulation of DSBs during neural development suggests that neurons possess mechanisms to cope with DSBs, which may even have a particular function or be generated as a side-effect of another, as-yet-unknown process. 

## 3. DSBs and Early Neural Cell Death

Several studies have shown that the number of neurons with somatic mutations decreases after development; both the number of CNVs per neuron [3] and the number of aneuploid neurons [67,68,69] are lower in the newborn and young adult brain than during embryonic development [9]. These observations suggest the involvement of cell death processes, which may selectively target non-viable mutations in a manner analogous to apoptosis following V(D)J recombination [70]. In fact, many authors, ourselves included, have shown that neural precursors and newborn neurons actually undergo a specific wave of apoptosis during early embryonic development [35,71,72,73,74]. Although the purpose of this early wave of neuronal cell death has not been clearly established, the findings in mutant models that lack apoptosis genes have underscored its importance. The dysregulation of cell death in this context specifically impairs proper CNS generation, affecting neural precursor cell proliferation and early neuronal differentiation [15,71,75]. Specifically, the dysregulation of cell death results in neural malformations including cerebral hyperplasia, exencephaly, and neural tube defects, as well as defective retinal structure and visual system connectivity [15,35,71,76,77,78]. 

Our work has shown that early neuronal cell death in the mouse retina parallels the onset of DSB generation (Figure 2), and that apoptosis occurs during a specific time window [48]. In the developing chicken retina, the onset of cell death is carefully programmed and intrinsically determined, specifically the death of newborn neurons is determined, in a cell-autonomous manner, by the time the neuron is generated and it is independent of the niche in which it was situated [79,80]. We and others have proposed that this may constitute a mechanism to eliminate abnormal, defective, or genetically unstable cells, thereby ensuring the selection of the fittest young neurons [67,73,81,82,83,84]. 

In neural development, DSB repair is a key step in controlling DNA rearrangements during neurogenesis, the failure of which results in programmed cell death [33,86,87]. Several findings suggest a close association between neuronal cell death and NHEJ, the main DSB repair pathway in neurons. Mice with impaired NHEJ show a dramatic increase in programmed cell death in neural tissues, frequently resulting in embryonic or perinatal lethality (Table 1). The phenotypes found in NHEJ repair mutants suggest that DSB generation and repair significantly influence the dynamics of neural development to promote neural diversity. Mice deficient in NHEJ proteins, such as KU-86, DNA polymerase mu, and DNA-PK, share a mild retinal phenotype characterized by moderate neuronal cell death ([39,40,61], Figure 2). Our studies in the developing retina of DNA polymerase mu- and DNA-PK-deficient mice have revealed a cell-autonomous phenotype of aberrant axonal navigation, similar to the phenotype observed in the cerebral cortex of NHEJ-1 mutants ([41,49,62]; Figure 2). This cell-autonomous axonal phenotype affects proteins involved in axonal structure and pathfinding (e.g., tubulin β3; Figure 2), as well as other proteins, such as the early neural genes BRAVO/NrCAM and L1CAM [49,62], supporting a possible role of DSB repair in axonogenesis. 

## 4. Mechanisms Underlying Specific DSBs: A Potential role for RAG-1,2 in Neural Development?

The abundance of DSBs created during neural development and the proposed requirement of neuronal diversity for correct development suggest the existence of specific mechanisms to promote neuronal diversity, reminiscent of those of the immune system. LINE-1 transposition is the only mechanism known to produce somatic DNA alterations in neural tissue during embryonic development, and has already been extensively reviewed [25,88,89]. Another two recently proposed DSB-generating mechanisms in the adult brain involve TOP2β, which is implicated in gene induction and chromosomal translocations elsewhere [90,91] and RNA retrotransposition, which results in exon shuffling in the APP gene [27].

A visionary hypothesis to explain the generation of diversity in the CNS pointed to the same mechanism that mediates V(D)J recombination in the immune system [92,93,94,95]. In developing lymphocytes, V(D)J recombination generates a highly diverse repertoire of antibodies via a process involving RAG-1,2 endonuclease. RAG-1,2 endonuclease activity is directed by canonical sequence-specific targets (recombination signal sequences, RSSs), which are later repaired via the NHEJ pathway [96,97]. Potential targets of RAG endonuclease (RSSs) are highly abundant in mammalian genomes (it has been suggested that there is a minimum of 10 million joining-sites in the genome [98]), although regulatory mechanisms prevent off-target activity [99]. Interestingly, a novel RAG-2 promoter has been described in diverse non-lymphoid tissues [100], and RAG-1 and RAG-2 expression in the nervous system has been described in the zebra fish olfactory bulb [101], mammalian retina [49,102], brain cell lines [103], and murine brain extracts [100].

In addition to the mild neural phenotype of RAG-1 and RAG-2 knockout models, there are further clues of a possible physiological role of RAG-1,2 in the nervous system. In humans, progressive encephalopathy has been associated with RAG-1 deficiency [104], while RAG-1-deficient mutant mice show defects in memory formation [105] and alterations in the olfactory system [106]. Experimental induction of RAG-1 also induces optic neuropathy, specifically increasing retinal ganglion cell (RGC) death [107]. Interestingly, our previous work showed that RAG-2 deficiency also impacts early retinal development (E13-E15), as reflected by a 23% decrease in DSBs, a 40% increase in early cell death, and cell autonomous axon guidance defects ([49]; Figure 2). 

The retinal phenotype of RAG-2 deficient mice closely resembles those of NHEJ DSB-repair mutants deficient in KU-86, DNA polymerase mu and DNA-PK [48,49,62,108], and also the cerebral cortex phenotype of the NHEJ-1 knockout mice [41], except for the reduction in the number of DSBs in the RAG-2 knockout.

The reduction in DSB number in RAG-2 deficient mice implies that RAG-2-endonuclease-dependent DSB generation and DSB repair may be active during early neuronal retinal development, and suggests that RAG-1,2 endonuclease participates in RGC neurogenesis, because its absence increases cell death of recently born neurons [49]. This fact, together with the observation that DNA polymerase mu deficient mutants may accumulate DSBs without terminal micro-homology [62], such as those mediated by RAG-1,2 [109], primed us and others to speculate a role for RAG-1,2 endonuclease activity coupled to NHEJ DNA repair mechanisms in the developing nervous system [49,95].

Several findings in the developing mouse retina reveal that RAG-2 deficiency also alters the expression of axonal proteins, as also described in SCID and DNA polymerase mu DNA repair mutants [48,49,62]. In RAG-2-deficient mice, we observed altered expression of at least four proteins implicated in axonal cytoskeletal structure and function, including tubulin β3, tubulin α 1C, fascin, and platelet-activating factor acetyl hydrolase 1 (Pafah1, formerly known as Lissencephaly-1). All genes encoding these proteins harbor putative RSSs that can be recognized by RAG-1,2 endonuclease, according to the RSS site database [49,110]. In summary, the available evidence indicates that two mouse models with mutations in NHEJ DNA repair proteins (DNA-PK and DNA polymerase mu) and the RAG-2 deficient mouse share impaired retinal ganglion cell (RGC) axonogenesis, altered axon fasciculation ([48,49,62], Figure 2), delayed optic nerve crossing at the optic chiasm, and cell-autonomous disturbances in the axonal trajectories, both in vivo and ex vivo [48,49,62]. Our work in dissociated retinal cultures indicates that axonal guidance defects are cell autonomous in all three of these mutant mice ([48,49,62], Figure 2). Taken together, these findings support a possible direct role of RAG-2 in axonogenesis and, potentially, axonal guidance. Of course, altered axonal pathfinding could be an indirect consequence of alterations in DSB generation and repair in genes related to axonogenesis in just a few neurons. When axonal projection is randomly altered in the neurons that produce pioneer axons, these pathfinding defects are further amplified by the aberrant projection of subsequent axons for which pioneer axons provide a scaffold [111]. 

Another potential explanation for the retinal phenotype of RAG-2 mutant mice is that the genes involved in axonal pathfinding may be specifically affected by a decrease in RAG-1,2-mediated DSBs. This again points to a mechanism to promote neuronal diversity in the genes involved in axon formation. 

The more recent discovery that CNVs of <1 Mb, which are comparable in size to RAG-1,2-dependent rearrangements observed in the immune system, are the most abundant somatic mutations in neuronal genomes [9] gives indirect support to a role for RAG-1,2 endonuclease activity and DNA repair mechanisms in the developing nervous system. Moreover, during neurogenesis, recurrent DSB clusters occur in the genes involved in neural cell adhesion and synaptogenesis, suggesting a potential impact of regulated DSBs on neurodevelopment and neural functions [66]. This hypothesis, thus, provides a plausible alternative explanation for the high degree of somatic mosaicism detected in adult neurons. 

In conclusion, DSBs that give rise to somatic mosaicism in the brain are mainly formed during early neurogenesis, possibly as a consequence of a programmed mechanism. Apart from DSBs resulting from LINE-1 translocation and TOP2β activity, it is plausible to hypothesize that DSBs generated by RAG-1,2 endonuclease activity in coordination with NHEJ could play a key role in neuronal development. RAG-1,2 activity may specifically modify the genes involved in the axonal structure, thus affecting cell survival and gene expression in surviving cells and contributing to the generation of neuron diversity.

## Figures and Tables

**Figure 2 ijms-23-06449-f002:**
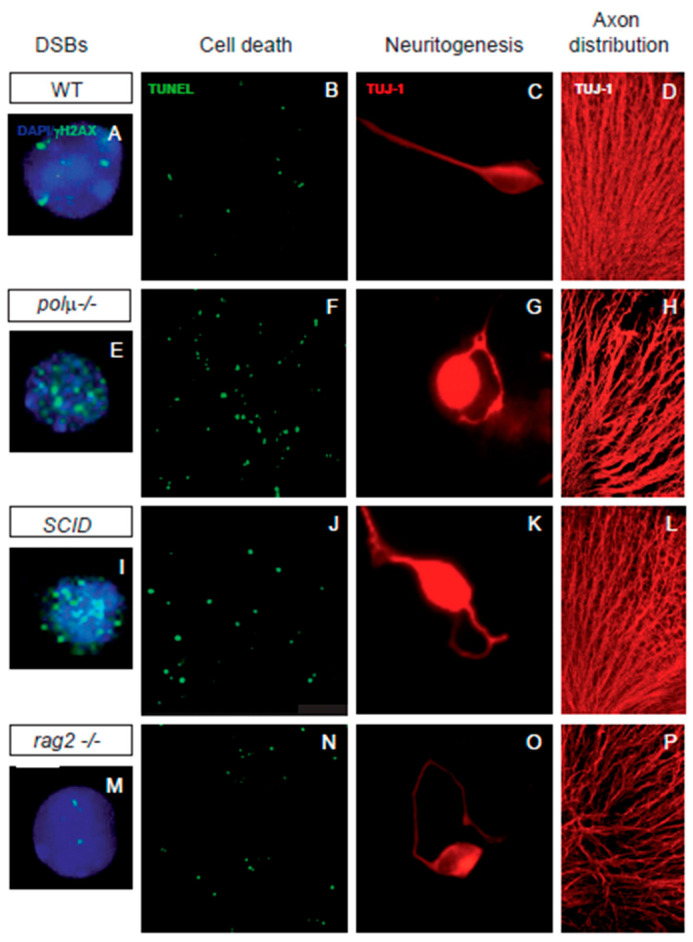
Impaired retinal development in mutant mice defective in components of DSB generation and repair mechanisms. Comparison of retinal phenotypes in mutant mice defective in DNA polymerase mu (polµ-/-; E-H), DNA-PK (SCID; I-L), and one of the subunits of the RAG-1,2 endonuclease responsible for generating the DSBs that originate the V(d)J recombination in the immune system (rag2-/-; M-P). The phenotype of the WT mouse is displayed in panels (**A**–**D,****A**,**E**,**I**,**M**) E13.5 dissociated retinal cells were immunostained for γH2AX (cyan) to reveal DSBs and counterstained with DAPI (blue) to visualize the nuclei. Notice that the foci numbers are increased in the repair defective mutants and reduced in the RAG2 defective mutant, with respect to the WT. (**B**,**F**,**J**,**N**) programmed cell death was detected by TUNEL (green) in whole mount E13.5 retinas. Notice that apoptotic nuclei numbers are increased in all three mutant mice with respect to the WT mouse. (**C**,**G**,**K**,**O**) E13.5 dissociated retinal cells were cultured on polyornithine/laminin-treated plates. Neurite emission was visualized by TUJ-1 immunostaining (red). Notice the disturbed axonal trajectories in all three mutant mice, with respect to the WT mouse. (**D**,**H**,**L**,**P**) E13.5 whole-mount retinas were immunostained with TUJ-1 (red) to visualize RGC axonal trajectories. Notice the disturbed axonal trajectories in all three mutant mice, with respect to the WT mouse. Images adapted from [48,49,62,85].

**Table 1 ijms-23-06449-t001:** Impact on nervous and immune system of defective proteins implicated in NHEJ. The table summarizes the phenotypes observed in human and murine mutants on proteins involved in NHEJ DSB repair.

NHEJ MUTATED GENE (and Function)	MURINE NEURAL PHENOTYPE	HUMAN PHENOTYPE Immune System and Genomic Stability	HUMAN PHENOTYPE Nervous System
LIG IV (DSB sealing)	Lethal in E14-E16 (depending on the study). Increased apoptosis in early postmitotic neurons. Acellularity in central and peripheral nervous system [15,37,38].	Immunodeficiency (residual T and B cells), pancytopenia, lymphomas, leukemia [36].	(Only hypomorphic mutants described). Microcephaly (non progressive after birth). Delayed development, primordial dwarfism and neurological abnormalities. Dubowitz syndrome, LIG4 syndrome [39].
Nhej-1/XLF/Cernunnos (DSB sealing)	Viable. Frequent spontaneous genomic instability, including translocations [40]. Increased neuronal cell death and neuronal migration defects in brain cortex [15,41].	Immunodeficiency (residual T and B cells), neutropenia, macrocytic anemia, autoimmunity [42,43].	In hypomorphic mutants, microcephaly, delayed development, chromosomal translocations. Nijmegen breakage syndrome-like phenotype, polymicrogyria [39].
XRCC4 (DSB sealing)	Lethal in E14,5. Increased apoptosis in early postmitotic neurons, acellularity in central and peripheral nervous system [15].	Genomic instability, hypersensitivity to radiation and cancer predisposition [44].	Microcephaly and delayed development. [44] Primordial dwarfism [45].
Pol β (DSB gap filling))	Neonatal lethality. Increased apoptosis in early postmitotic neurons, apoptosis in central and peripheral nervous system, genomic instability [15].	Genomic instability [46].	Reduced activity in patients with Alzheimer disease [47].
DNA-PK (Nuclease. DSB end processing)	Viable. Increased apoptosis in early retina postmitotic neurons [48]. Altered axonal emission [49]. If combined with Pol β deficiency, lethal in E11,5, delayed embryonic development and massive neuronal apoptosis [15,50].	Severe combined immunodeficiency, total loss of T and B cells [51,52].	Microcephaly, delayed development, progressive neural degeneration and telomere shrinkage [39,53].
Artemis (Nuclease. DSB end processing)	Viable. Hypersensitivity to radiation and genomic instability, including telomeric fusions [54].	Progressive immunodeficiency, reaching total T and B cell loss, autoimmunity and Omenn Syndrome. Leukemia and non lymphoid carcinomas [39].	Not described.
MRE11/NBS1-1/RAD50 (Sensor of DNA damage)	Lethal at E6. Elevated genomic instability [55,56].	Predisposition to lymphomas, breast and ovary cancer [39].	Nijmegen breakage syndrome (NBS), microcephaly and ataxia [39].
KU 70/80 (Recognition of DNA lesions)	Viable. Increased apoptosis in early postmitotic neurons, especially in the retina [57].	Suspected to induce embryonic lethality due to telomeric instability [58].	Melanoma brain metastases with high genomic instability [59].
ATM (Sensor of DNA damage)	Viable. Delayed embryonic development, with neurologic disfunction [15]. Specific loss of a subpopulation of dopaminergic neurons [60]. Hypersensitivity to radiation. [61].	Reduced or absent levels of IgE, IgA and IgG2, genomic instability, telomere shrinkage and lymphoma predisposition [39].	Ataxia, progressive neurodegeneration, ocular telangiectasia [39].
Polymerase mu (DSB gap filling)	Viable. Increased apoptosis in early retina postmitotic neurons, ectopic neurons and axonal pathfinding cues, and altered axonal emission [62]. Increased learning and brain long term potentiation in aged mice [63].	Not described in humans, but altered hematopoiesis has been detected in mice [64].	Not described.

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
