# Peer review of "Exploring the Origin and Physiological Significance of DNA Double Strand Breaks in the Developing Neuroretina"

_ijms, 2022, doi:10.3390/ijms23126449_

Round 1
Reviewer 1 Report
The article presented by the authors is interesting and original, well written and structured, however I have a major concern:
the authors presented in figure 2 different immunofluorescence on mice tissue, but it is not clear how they performed the experiment and they should present an ethical permit.
The authors should insert a material and method section.
Author Response
Reviewer 1
Thanks for your comments on the manuscript.
Regarding your major concern:" the authors presented in figure 2 different immunofluorescence on mice tissue, but it is not clear how they performed the experiment and they should present an ethical permit.
The authors should insert a material and method section."
All panels in Figure 2 have been taken, original or adapted, from our previous publications, as in indicated in the figure legend. However, we have revised the MS and made little changes to clearly state that the figure shows already published work and of course it has been performed with all ethical permits, as our institution requires.
This article is a review, which usually does not include a M&M section. However, in the case that you consider it necessary, we can certainly include M&M as supplementary material.
Reviewer 2 Report
The paper entitled “Exploring the origin and physiological significance of DNA double-strand breaks in the developing neuroretina.” is a study based on the role of genetic variations in normal neural development and its possible impact on brain diseases. The study also reports the relationship between DNA breaks, neurogenesis, and early neuronal cell death in the developing retina.
The review is well written and of clinical interest, especially the section regarding DNA breaks that give rise to somatic mosaicism in the brain, which are formed during early neurogenesis, possibly as a consequence of programmed mechanisms.
The role of factors that influence neuroretina development are useful in understanding physiopathological pathways and can assist in paving the way for future studies in the field of sight-threatening diseases.
The study has been correctly planned. The study provides possible mechanisms during neuroretina development. Minor modifications and polishing can improve the flow of the paper.
Author Response
Thanks for your comments on our work.
About your concern: "Minor modifications and polishing can improve the flow of the paper."
Our paper had been corrected by a native English PhD and we have now tried to improve the flow of the MS with minor changes by a native English-speaking colleague.
Round 2
Reviewer 1 Report
The author satisfied all my concern. I propose the acceptance in the present form.